# Towards time privacy policies in ODRL

Juan Cano-Benito[1], Andrea Cimmino[1] and Raúl García-Castro[1]

[1]*Ontology Engineering Group, Universidad Politécnica de Madrid, Madrid, Spain*

## Abstract

The Open Digital Rights Language (ODRL) is a standard widely adopted to express privacy policies, but presents some challenges at the ontology level, specification, expressiveness, and privacy policy nature when solving queries. The article presents that some challenges should be addressed by extending the ODRL ontology, aligning it to other well-know ontologies, expanding the ODRL ontology to support time policies properly. Next, different examples of time-based ODRL policies implemented with a developed enforcement engine are presented below. Finally, conclusions and future work are presented.

## Keywords

ODRL, privacy policy, ontology, enforcement

## 1. Introduction

In recent years, the use of digital devices has increased, leading in particular to a huge increase in the number of IoT devices deployed with a multitude of domains of diverse nature such as home automation, industrial, medical, military, among others[1]. These IoT devices and services are commercial products that provide access to their data in different protocols (MQTT, HTTP, CoAP...) and expressed in different formats (JSON, CSV, SQL..,) and even if the data is expressed in the same format, it is possible to provide different data models.

These digital devices are widely used in dataspaces, which are ecosystems where users voluntarily contribute with data from their devices, where part of the data can be generated by IoT devices [2, 3]. Due to the heterogeneity of IoT devices, dataspaces have to deal with different data sources, and the solution to this heterogeneity is to establish a semantic interoperability framework.

Once this interoperability between the different devices is achieved, the data can be accessed from any other device within the dataspace. However, this information provided by the devices can be highly sensitive [4], so dataspaces can incorporate security measures to access these devices. Despite these measures, users may wish to incorporate certain privacy policies that do not address the security measures of the dataspaces, such as access to devices under certain conditions (e.g. access only at certain times of the day to their devices or based on their geographic location).

These privacy policies can be expressed through the Open Digital Rights Language (ODRL) language [5]. ODRL is a W3C standard ontology that provides the vocabulary to describe

*NXDG 2024: NeXt-generation Data Governance workshop, September 17, 2024, Amsterdam, Netherlands*

✉ juan.cano@upm.es (J. Cano-Benito); andreajesus.cimmino@upm.es (A. Cimmino); r.garcia@upm.es (R. García-Castro)

🆔 0000-0002-5638-4977 (J. Cano-Benito); 0000-0002-1823-4484 (A. Cimmino); 0000-0002-0421-452X (R. García-Castro)

policies in decentralised ecosystems, such as dataspaces (ODRL is currently being used by the International Data Space Association[1] or dataspaces projects such as Gaia-x [6]), providing a privacy data layer. Although ODRL is focused on representing statements about the use of content and services, the ODRL language is aimed at legal domains and therefore has limitations in its vocabulary for representing other policies outside of that domain [7] and other technical limitations [8], such as the lack of implementations or the limitations in expressiveness.

Therefore, this paper presents an extension of the ODRL ontology to represent temporal policies, extending the scope of ODRL privacy policies with time constraints, thus providing a way to extend the range of policies that can be applied with ODRL. In addition, examples of use cases that use time-constrained ODRL policies and examples using an enforcement engine to enforce ODRL policies is shown.

The rest of the paper is organised as follows: Section 2 analyses the related work done to expand the ODRL privacy language; Section 3 explains how to use ODRL and summarises the main limitations of this policy language; Section 4 presents the extension of the ODRL ontology to provide time privacy policies; Section 5 shows use cases and how privacy policies have been implemented using a enforcement engine; finally, Section 6 recaps our conclusions and main findings.

## 2. Related work

The analysis of privacy policies for data sharing has been extensively analysed by researchers [9] and the security and privacy of geospatial data is an important known topic for researchers and industry [10]. In the field of the ODRL language, there is an effort to expand this policy language [11, 12, 13, 14, 15, 16] to support more functionalities.

Steyskal and Polleres [11] address the use of more expressive and detailed access policies for Linked Data than is possible with ODRL; which was originally intended to be used to define an open standard for policy expressions for digital media. Later, Steyskal and Polleres [12] discuss how to improve ODRL privacy policies, proposing a methodology to manage dependencies between actions.

Kim and Chung [13] present an extended model of ODRL to manage copyright on user-generated content, allowing users to specify and control licences to distribute their digital assets. De Vos et al [14] propose a model to verify the compliance of business processes with regulatory obligations, using examples where the General Data Protection Regulation (GDPR) is used to illustrate the functionality of the proposed model. Finally, Esteves et al [15] extends the access control in Solid using ODRL, allowing more complex privacy policies, but do not extend the ODRL ontology to allow for time privacy policies.

These works based on ODRL do not extend the ontology or cover time privacy policies. The only work that covers time privacy policies is the work of Akachi et al [16]. However, in this work, the operators used are the ODRL operators, which would not allow time comparisons. Then, in this paper presents an extension of ODRL with temporal ontologies and examples of how to implement this time-based privacy policies, and run privacy policies implemented in a developed enforcement engine.

---

[1]https://github.com/International-Data-Spaces-Association/IDS-RAM_4_0

# 3. Using ODRL

ODRL is an W3C open standard designed to express and communicate digital rights policies in a standardised way, providing a common language that allows users to define terms of use for data. ODRL allows to possible to specify permissions, prohibitions and duties related to the use of digital content. Furthermore, being an open standard, this facilitates interoperability and flexibility when combined with other ontologies, allowing for a wider range of constraints.

The ODRL ontology (depicted in Figure 1) is the formal implementation of this language in a machine-readable format. This ontology translates ODRL concepts (permissions, prohibitions and duties) into classes and properties that can be used in the context of the semantic web. In order to extend the functionalities of ODRL, this work will extend the Operator class.

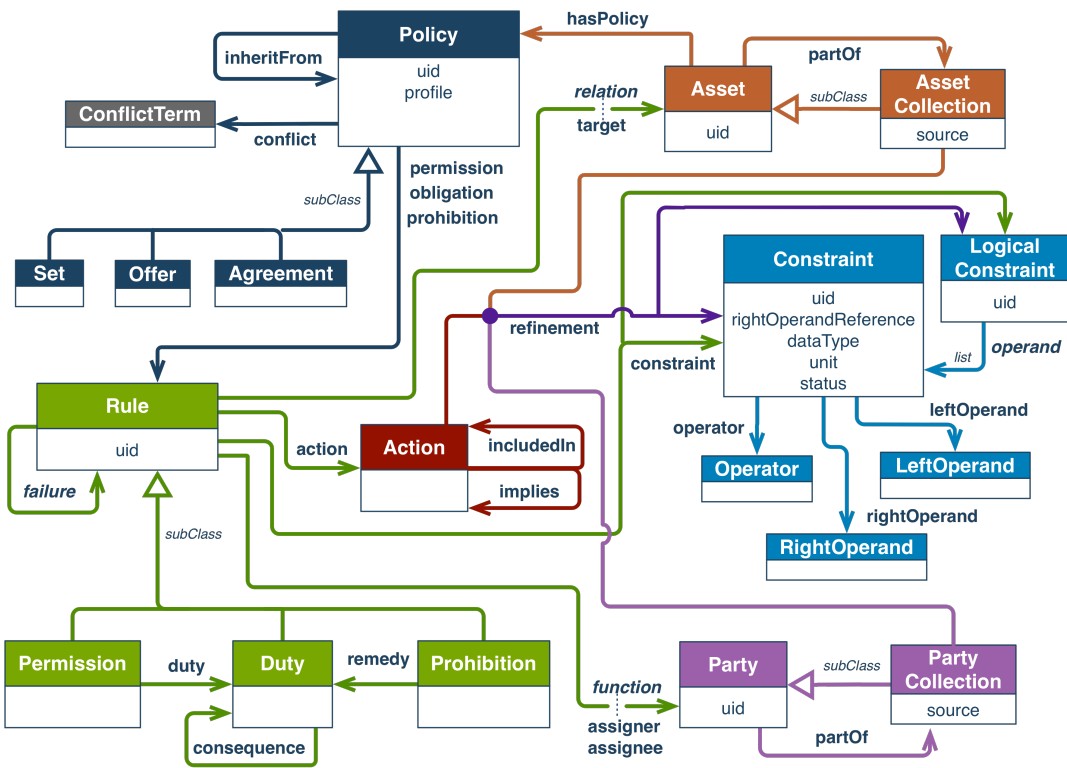

**Figure 1:** ODRL ontology

Using the semantic web and using the ontology, different examples of an ODRL policy expressed using the JSON-LD 1.1 syntax, which is a form of serialisation for RDF graphs commonly used in the context of the semantic web, can be constructed (Listing 1), as show in the example 13 of the ODRL documentation[2].

---

[2]https://www.w3.org/TR/odrl-model/#constraint-rule

Listing 1: Example of ODRL privacy policy

```json
{
    "@context": "http://www.w3.org/ns/odrl.jsonld",
    "@type": "Offer",
    "uid": "http://example.com/policy:9090",
    "profile": "http://example.com/odrl:profile:07",
    "permission": [{
        "target": "http://example.com/game:9090",
        "assigner": "http://example.com/org:xyz",
        "action": "play",
        "constraint": [{
            "leftOperand": "dateTime",
            "operator": "lt",
            "rightOperand": { "@value": "2017-12-31", "@type": "xsd:date" }
        }]
    }]
}
```

This policy allows to play one game if the date is less than 31 December 2017. Looking at this time-based ODRL privacy policy and the ODRL documentation, certain observations in ODRL can be identified.

- **Ontological level.** In the ODRL ontology, to measure times, the equals of the ODRL ontology are not the same as those used in other time ontologies, and the Greather Than and Less Than of the ontology are not specified to handle dates, so the ODRL ontology must align with other ontologies to be semantically correct. Therefore, there is no alignment of the semantics of the ODRL logical operators.
- **Lack of implementation specification.** ODRL does not provide an implementation specification for the different operands and operators, and it should be assumed that all the *LeftOperand* and *RightOperand* implementations must provide data that come from inside a system and is somehow accessed during the evaluation of the policy.
- **Limitations in the expressiveness.** The ODRL specification is not detailed enough on how the policies are evaluated by a software system in order to grant or revoke an action over a target (i.e., enforcement).

These observations leads to enforcement engine limitations. At the ontological level, precise interpretation of policies is made problematic. The absence of a detailed implementation specification for operands and operators can lead to inconsistencies in policy enforcement between different systems, leading to semantic interoperability problems (two different ODRL privacy policies follows different ontologies). In addition, the lack of clarity in the specification of how policies should be evaluated and the binary nature of ODRL limits the ability of engines to handle complex scenarios.

## 4. Extending ODRL with time

This section aims to extend the ODRL ontology provided with the main time relations to enable ODRL policies to be correct from a semantic web point of view. There are many ontologies

dealing with time, however, in this paper we have used the OWL-time ontology, which is currently a candidate for W3C recommendation [17] and has been verified in previous work [18].

The ontology presented in this section was built following the LOT (Linked Open Terms) methodology [19]. This methodology, based on agile techniques, is composed of a four-stage workflow: requirement specification, implementation, publication, and maintenance.

- **Ontology requirements specification.** To capture the requirements, the ODRL ontology and different time-entities and geographical ontologies were studied.
- **Ontology implementation.** This phase was split into three sub-tasks. Firstly, a conceptualisation of the basic concepts of ODRL[3] and time[4] ontology were extracted, and the relationships between these concepts were identified from the ODRL and time documentation. Secondly, in the encoding step, a model was generated in the OWL language from the ontological model using Protégé. Finally, a group of experts supervised the ontology to check that it has no syntactic, modelling, or semantic errors and complies with all the requirements captured in the previous phase.
- **Ontology publication.** The ontology is published in a GitHub repository[5]. The ontology include the code in OWL, a human-friendly documentation with a description of the classes, properties and data properties, and a graphical representation of the ontology.
- **Ontology maintenance.** The ontology will be updated to correct possible errors or implement new requirements that will be incorporated in future versions of ODRL.

This extension of ODRL has been made in the context of the AURORAL project, a digital service platform tailored to the needs of rural communities. These ontologies and their elements are identified using Internationalized Resource Identifiers (IRIs). For the sake of simplicity, in the rest of this article, these IRIs will be referenced using the prefixes defined in Listing 2.

Listing 2: Predefined namespace prefixes

```
time: http://www.w3.org/2006/time#
odrl: http://www.w3.org/ns/odrl/2/
```

For the OWL-time there are 13 classes that define relations between time periods, and the result of these relations are binary, in line with ODRL policies. Then, it is possible to extend the ODRL ontology with this ontology in order to solve privacy policies based on time with the appropriate operator. The resulting ontology is depicted in Figure 2. These classes defining the relationships between two time periods have the normal relationship and the inverse and are explained below:

- **Before and After.** A direction in time is assumed. In *"Before"*, if a time entity T1 is before another time entity T2, then the end of T1 is before the beginning of T2. *"After"* is the inverse.
- **Meets and MetBy.** If a proper interval T1 is *"Meets"* another proper interval T2, then the end of T1 is coincident with the beginning of T2. *"MetBy"* is the inverse.

---

[3]https://www.w3.org/TR/odrl-model
[4]https://www.w3.org/TR/owl-time
[5]https://github.com/ODRE-Framework/odre-time

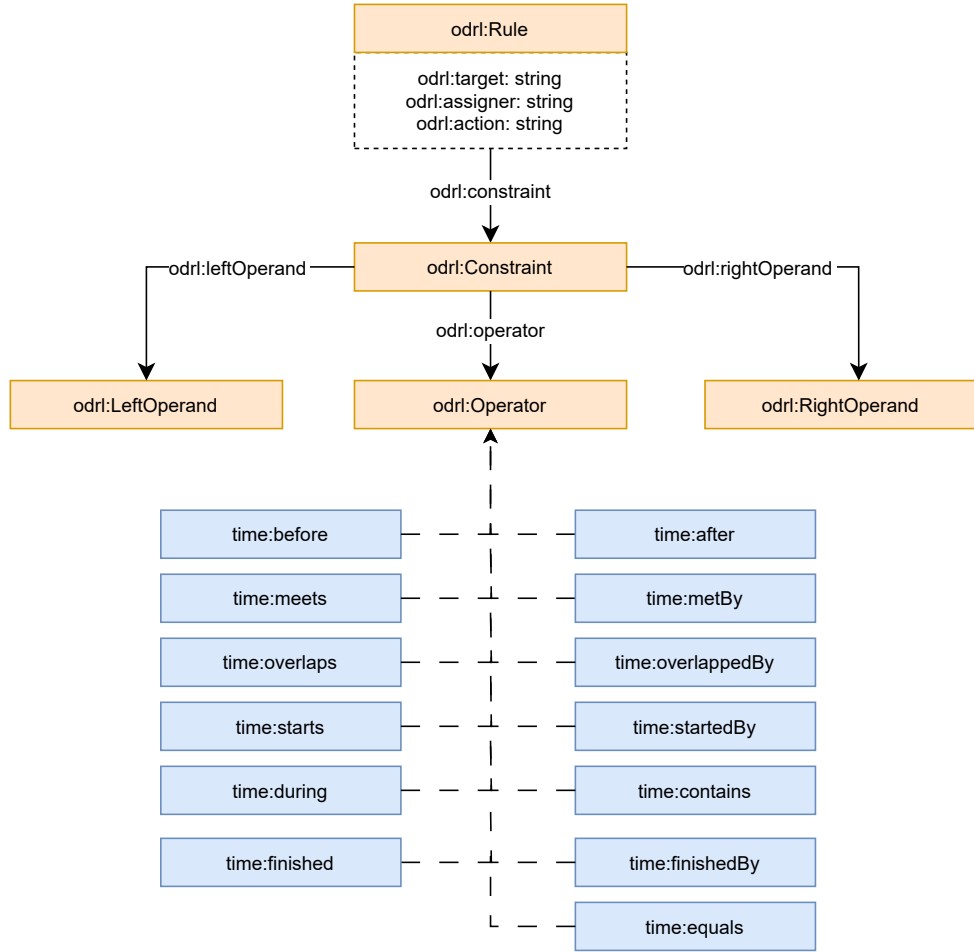

**Figure 2:** ODRL extension with OWL-Time ontology

- **Overlaps and OverlappedBy.** If a proper interval T1 is *"Overlaps"* another proper interval T2, then the beginning of T1 is before the beginning of T2, the end of T1 is after the beginning of T2, and the end of T1 is before the end of T2. *"OverlappedBy"* is the inverse.
- **Starts and StartedBy.** If a proper interval T1 is *"Starts"* another proper interval T2, then the beginning of T1 is coincident with the beginning of T2, and the end of T1 is before the end of T2. *"StartedBy"* is the inverse.
- **During and Contains.** If a proper interval T1 is *"During"* another proper interval T2, then the beginning of T1 is after the beginning of T2, and the end of T1 is before the end of T2. *"Contains"* is the inverse.
- **Finished and FinishedBy.** If a proper interval T1 is *"Finished"* another proper interval T2, then the beginning of T1 is after the beginning of T2, and the end of T1 is coincident with the end of T2. *"FinishedBy"* is the inverse.
- **Equals.** If a proper interval T1 is *"Equals"* another proper interval T2, then the beginning

of T1 is coincident with the beginning of T2, and the end of T1 is coincident with the end of T2.

## 5. Time ODRL policies

To address the ontological limitations of ODRL, specifically in the handling of temporal concepts, an example of a privacy policy is shown to demonstrate the integration of the ODRL ontology with the OWL-Time ontology. This combination aims to improve the expressiveness and accuracy of privacy policies in scenarios that require temporal representation. Listing 3 shows an ODRL privacy policy using in the *operator* value a class of the OWL-Time ontology.

Listing 3: Example of ODRL privacy policy with ODRL time

```
{
    "@context": "http://www.w3.org/ns/odrl.jsonld",
    "@type": "Offer",
    "uid": "http://example.com/policy:9090",
    "profile": "http://example.com/odrl:profile:07",
    "permission": [{
        "target": "http://example.com/game:9090",
        "assigner": "http://example.com/org:xyz",
        "action": "play",
        "constraint": [{
            "leftOperand": "@value": "2017-12-31", "@type": "xsd:date" ,
            "operator": "time:before",
            "rightOperand": { "@value": "2019-12-31", "@type": "xsd:date" }
        }]
    }]
}
```

Listing 4 shows an example of an ODRL policy with time values. In this policy, two dates are evaluated and the *time:after* operator is used to check if the date in the *leftOperand* operand is older than the date in the *rightOperand* operand.

Listing 4: Example of ODRL privacy policy with ODRL time

```
{
    "@context": "http://www.w3.org/ns/odrl.jsonld",
    "@type": "Offer",
    "uid": "http://example.com/policy:6163",
    "profile": "http://example.com/odrl:profile:10",
    "permission": [{
        "target": "http://example.com/document:1234",
        "assigner": "http://example.com/org:616",
        "action": "distribute",
        "constraint": [{
            "leftOperand": { "@value": "2019-12-31", "@type": "xsd:date" },
            "operator": "time:after",
            "rightOperand":  { "@value": "2017-12-31", "@type": "xsd:date" }
        }]
    }]
}
```

Enforcement engines are software components that enable the execution of privacy policies expressed in ODRL. These engines interpret, evaluate, and enforce the rules defined in ODRL policies, translating from ODRL expressions into enforceable rules, to evaluating conditions, and making decisions on whether a specific action is allowed or prohibited, based on the current policy and context. However, relying only on the ODRL documentation, these engines may have limitations such as interpreting complex policies or handling scenarios that require external information. In order to overcome these limitations, in addition to extending the ODRL ontology, an enforcement engine is used[6] that can implement more complex policies.

This enforcement engine used can implement ODRL privacy policies with external data. Listing 5 shows a privacy policy in which the current system time is being added to the privacy policy. With this enforcement engine, using *"datetime"* in value, the datetime of the system is retrieved.

Listing 5: Example of ODRL privacy policy with ODRL time and external data in a enforcement engine

```
policy = """
{
    "@context": "http://www.w3.org/ns/odrl.jsonld",
    "@type": "Offer",
    "uid": "http://example.com/policy:6163",
    "profile": "http://example.com/odrl:profile:10",
    "permission": [{
        "target": "http://example.com/document:1234",
        "assigner": "http://example.com/org:616",
        "action": "distribute",
        "constraint": [{
            "leftOperand": "dateTime",
            "operator": "time:equals",
            "rightOperand":  { "@value": "2024-10-22", "@type": "xsd:date" }
        }]
    }]
}
"""
usage_decision = ODRE().enforce(policy)
print(usage_decision)
```

## 6. Conclusions and future work

Although the ODRL recommendation has been widely adopted, ODRL itself has certain limitations, such as at the ontological level, specification, expressiveness, and the binary nature of ODRL when solving queries, making it difficult to consolidate ODRL as a privacy policy language. As a solution, this article proves how to overcome the limitation comparing different times in ODRL with the correct operator by aligning the ODRL ontology to the OWL-Time ontology, and building and testing an enforcement engine to solve privacy policy queries. Future work will consist of extending the ODRL privacy policies with other ontologies and adding more features to the ODRL enforcement engine.

---

[6]https://pypi.org/project/pyodre

## Acknowledgments

This work is partially funded by the European Union's Horizon 2020 Research and Innovation Programme through the AURORAL project, Grant Agreement No. 101016854 and by the Madrid Government (Comunidad de Madrid-Spain) under the Multiannual Agreement with the Universidad Politécnica de Madrid in the Excellence Programme for University Teaching Staff, in the context of the V PRICIT (Regional Programme of Research and Technological Innovation).

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
