# OpenReview forum: "Towards time privacy policies in ODRL"
_SEMANTiCS.cc/2024/Workshop/NXDG — NXDG 2024_

### Official Review · ~Rui_Zhao15 · 2024-07-30
**Sensible solution for temporal constraints in ODRL, with some other remarks**

**Rating:** 7
**Confidence:** 4

**Review:**

This paper presents an extension to ODRL, aiming to address the lack of clear semantics of temporal constraints. It uses the OWL-Time ontology to define a new set of operators for temporal relations.

The problem of temporal constraints and the reason to use OWL-Time ontology is clearly stated, and the paper is self-contained to have introduced all necessary concepts of the introduced ontology, together with some examples; the approach is sensible and addresses the target problem. But there are some typos or misrepresented information which should be addressed in the final version.

1. On page 6, for the list of introduced operators, the sentences don't read fluent enough, because the operators are presented as verbs (third-party tense) while the sentence also has its own verb. Also, the inverse is sometimes presented with upper case and sometimes doesn't.
2. The last sentence before Listing 3 seems to have typos, leading to problem when reading.
3. Listing 4 uses `time:after`, but the explanatory paragraph says `time:before`.
4. For then sentence before Listing 5, I'm having some difficulty in understanding it. The example only added curly brackets around a magic date `2024-01-01`. How does that denote the current system time?
Apart from this, it is not clear how this makes the policy more interoperable, which seems to be the issue that introducing this engine aims to overcome? Could the authors elaborate more?
In particular, by reading the paragraph above, it seems the issue is: contextual (external) information may be needed to evaluate some policies, but the method/mechanism to retrieve such information is not defined by the ODRL documentation. So presumably the engine should provide a standard and interoperable way to retrieve such information (which is not shown here)? Maybe I misunderstood?
5. For Listing 1, the authors discussed the mismatched type of `dateTime` and `xsd:date`, while for Listing 3 they used `xsd:dateTime` altogether. An explanation is needed for the design consideration, e.g. why using `xsd:dateTime` for a seemingly date-only information, and how that interoperates with `xsd:date` (if that is allowed).
6. There is a period (`.`) on line 3 of Sec 6, which seems to be a grammar mistake.
7. In the next sentence (starting in line 4 of Sec 6), it is not clear which "limitation" it is talking about -- not all limitations (listed in the previous sentence) are addressed in this paper.

---

### Official Review · ~Wout_Slabbinck1 · 2024-08-01
**An interesting paper for the NeXt-generation Data Governance workshop about ODRL temporal policies and an ODRL enforcement engine**

**Rating:** 6
**Confidence:** 4

**Review:**

The authors give a relevant paper that demonstrates an ODRL enforcement engine that works with their proposed extension to ODRL to support time policies.

In the abstract, the authors express the need to extend ODRL to support time policies properly.
Furthermore, it states that in this work they tackled this challenge by extending ODRL to support time policies and that they have implemented an ODRL engine to support some ODRL time examples.
The abstract further mentions that conclusions and future work are presented in the paper, but a small summary in the abstract of the content of the conclusion and future work is lacking.

Overall, it is a clear and concise abstract.

In the next couple of paragraphs, some comments are provided which, when them taking into account, might improve the paper.

In the introduction, the authors provide some context with regarding to dataspaces with following sentence: "These digital devices are widely used in dataspaces, which are ecosystems where users voluntarily contribute with data from their devices, where part of the data can be generated by IoT devices"
Looking at the cited source, the above sentence is not entirely clear as:
(i) the paper cited does not mention voluntary contributions of users with data, and
(ii) the cited paper states that dataspace is still in its infancy (see discussion section) and that "Unfortunately, the number of dataspace applications in business is limited to a few" indicating that even if iot devices are widely used in dataspaces, that does not mean too much.

As a suggestion, it might be more clear if an interpretation/definition of dataspaces is added that further strengthens your point (such as [1], which gives a good overview of the origins of dataspaces and mentions several definitions).

In dataspaces, the need for privacy policies due to sensitive data is well motivated.
A link to ODRL as privacy policy is given, but maybe some citation to Gaia-x [2] and IDSA [3] (dataspace organisation) that do mention the fact ODRL is used might make the use of ODRL more clear.

When talking about the limitations of ODRL " and other technical limitations" is mentioned. Could an example be given in the introduction such that it is clear for the reader what some of the pain points actually are (or the most concrete one)?

You mention in related work that the papers do not extend the ODRL ontology or do not cover time privacy policies. However, in SDS24 (https://dbis.rwth-aachen.de/SDS24/), a paper [4] was presented that does cover dynamic policies with a suggestion for extension of ODRL that could support time-based ODRL policies (see section 4 Dynamic ODRL Specification ). This indicates that there are papers that extend the ODRL ontology and do cover temporal privacy policies.

In "Using ODRL"  it is not clear what following sentence means. It is suggested to rewrite it such that it becomes more clear to the reader:
"For the purpose of extend the functionalities of ODRL, the ODRL ontology (depicted in Figure 1) is the formal implementation of this language in a machine-readable format."
Furthermore, he full ODRL ontology is added as a figure, but no additional information on why or other references apart from the fact that it is the ontology is provided. My suggestion is to remove this altogether as the introduction of ODRL speaks for itself.

In the sentence that introduces Listing 1, a reference is being made to example 13 of ODRL. However, it is example 18 that is being shown as a listing (https://www.w3.org/TR/odrl-model/#permission).
If this was intentional, it would become more clear if the authors added a description on why example 18 was shown and not 13 and why this particular example is so important.

Finally, in the "Using ODRL" section, the authors point out several observations in ODRL.
On the ontological level, they claim that ODRL is domain specific.
Though, the ODRL Information model describes ODRL as a flexible and interoperable model for representing statements of usage of content and services.
The way it is written is rather generic, so arguments can be made that ODRL is rather domain agnostic.
A suggestion to the authors is to keep the observation, but remove the notion of domain specific and rather focus more on the argument of no alignment of the semantics of the ODRL logical operators.
This argument makes sense as currently, there still is not a complete formal semantic description of the evaluation of ODRL policies, though work has started on formalisation in the ODRL formal semantics CG (https://w3c.github.io/odrl/formal-semantics/).

Furthermore, I would like to point out that, while the "binary nature" observation is correct, I'd argue that it does not limit the creation of complex queries over ODRL Policies.
An evaluation of ODRL policy would be against some context (which could be a request, time, ...). At the end of such an evaluation, you would want a yes or no such that in an access control enforcement system you could then grant or deny access to a given target resource.
When the context changes (e.g. the time is now different), a different evaluation result could be present.
Depending on the context and the policy, this could be rather complex. So I would suggest the authors to either elaborate on why complex queries are not possible or either remove the fact that the binary nature would be insufficient for evaluation purposes.

Compliments on the authors for the sentence about the lack of detailed implementation specification. This is really powerful and indeed needs to be addressed in ODRL.

In section 4, the reason for choice of ontology and the methodology used for extending ODRL with time is well written and well motivated.

In the fifth section, while describing Listing 4 the `time:before `operator is mentioned. However, in the example itself the `time:after` predicate is used.
The description of Listing 4 seems to indicate that the former should be used, so this typo must be fixed.

Furthermore, it is not clear why two almost identical policies are shown. For readability, it would be better if one remained for a final version unless the need for distinction between the two is well motivated in the paper.

When talking about enforcement engines, the motivation and need is very clear. But, I'd suggest to mention that others have noticed this as well and try to formalise this in the ODRL formal semantics group.

A small note on Listing 5. I've tried to run it with `pyodre`, but it gave a couple of errors. The examples provided in the repository did run. Though, initially I was confused why next to the allowed permission, a `None` appeared. It might help to document the expected outcome in the open source repository.

A final comment, which might of great importance, is the fact that the authors do not follow the ODRL model regarding the constraints.
The range of `odrl:leftOperand`  must be of class `odrl:LeftOperand`. Instances such as `odrl:dateTime` or `odrl:count` are correct.
However, in a couple of examples the authors use as object in such a triple a value of type `xsd:dateTime`.
While it is clear why they do it, this would not interoperate with other ODRL policies that do follow the specification.
A suggestion to combat this is either to motivate the need for concrete values as left operands and the need for adopting this into the ODRL specification or to remove such concrete examples from the examples altogether.

Finally, there are some typos in the paper:
- Section 2 analyse should be Section 2 analyses
- In figure 2, there is a typo: `priv:LeftOperand` should be `odrl:LeftOperand`

To conclude, I think this is an interesting paper for the NeXt-generation Data Governance workshop. Though an iteration addressing the comments mentioned above is required.

[1] J. Theissen-Lipp et al., ‘Semantics in Dataspaces: Origin and Future Directions’, in Companion Proceedings of the ACM Web Conference 2023, Austin TX USA: ACM, Apr. 2023, pp. 1504–1507. doi: 10.1145/3543873.3587689.

[2] GAIA-x mentioning ODRL (end of 1.4): V. Siska, V. Karagiannis, and M. Drobics, ‘Building a Dataspace: Technical Overview’. Gaia-X Hub Austria. https://www. gaia-x. at/wpcontent/uploads/2023/04 …, 2023. Available: https://www.gaia-x.at/wp-content/uploads/2023/04/WhitepaperGaiaX.pdf

[3] IDS-RAM mentioning ODRL (4.1.6 Usage Control) : International-Data-Spaces-Association/IDS-RAM_4_0. (Dec. 18, 2023). International Data Spaces Association. Available: https://github.com/International-Data-Spaces-Association/IDS-RAM_4_0
https://github.com/International-Data-Spaces-Association/IDS-RAM_4_0/blob/1fcdec06a36296e322f66a5daf934f9617a4da4e/documentation/4_Perspectives_of_the_Reference_Architecture_Model/4_1_Security_Perspective/4_1_6_Usage_Control.md

[4] I. Akaichi et al., ‘Interoperable and Continuous Usage Control Enforcement in Dataspaces’, in Proceedings of the Second International Workshop on Semantics in Dataspaces (SDS 2024), J. Theissen-Lipp, P. Colpaert, S. K. Sowe, E. Curry, and S. Decker, Eds., in CEUR Workshop Proceedings, vol. 3705. Hersonissos, Greece: CEUR, May 2024, Available: https://ceur-ws.org/Vol-3705/#paper10

---

### Decision · Program_Chairs · 2024-08-02

Accept